# Dynamical Networks Modelling Applied to Low Voltage Lines with Nonlinear Filters

**Mauro Fazion Filho** [1,2]

1 School of Mathematics, University of Bristol, Bristol BS8 1TH, UK; mauro.fazion@bristol.ac.uk
2 Unisul Virtual, UNISUL Universidade do Sul de Santa Catarina, Palhoça 88137-900, Brazil

**Abstract:** The nonlinear dynamical behaviour of a network that is submitted to disturbances is the starting point of this work, where we consider a low voltage line (the network) with nonlinear varistor filters responding, dynamically, to those disturbances. Network models consider static measurements, and here we develop an iterative model to deal with dynamical measurements. We begin with the one-dimensional communication line model using reflected and incident signals, which are dependent on the node parameters, proceeding a time-step computation. Each node is a space representation that consolidates parameters for a specific vertex and its edges. Nonlinear functions are applied within the node and will contribute to the general process running on the structure. The idea of a structure and its related processes leads to a new concept of sustainability and system robustness. This concludes the work, along with several experimental and simulation results, with direct advantages to electromagnetic interference control and mitigation.

**Keywords:** network; dynamical network; nonlinear; transmission line method; electromagnetic compatibility; communication line; sustainable system

## 1. Introduction

### 1.1. Motivation and Contribution

The core motivation of this research comes from a reflection about network sustainability and robustness. We consider network robustness as a composition of two factors: the network physical structure (static) and the processes running (dynamical) on that structure. Currently, networks are analysed, mostly, from a static perspective. Vertices and edges build static networks from which several measurements and results are extracted. This approach works very well for many theoretical and practical problems. What happens, however, when those vertices and edges are changing, arising, disappearing and creating new relationships anytime? What if these changes are nonlinear? Are these networks still sustainable and robust during and after the nonlinear occurrences?

In a real-world system, we could get acceptable outcomes for a process even when its physical structure is severely damaged. For example, the transport and delivery of goods can happen despite roads being in bad condition. In other cases, the physical structure is fine but the processes are out of control and there are no outcomes, like when deliveries do not occur because of traffic jams, even if the road infrastructure is good.

In this work, we consider that the complementary relations of these two factors, (1) the physical structure and (2) the processes running on top of it, are responsible for the general system robustness level, i.e., its level of robustness (or sustainability) depends on how simultaneously robust its structure and its processes are. That level of robustness should be measured every and anytime, especially if nonlinear occurrences come to affect the system as a whole.

However, creating a general measurement for the system robustness level is not an easy task and it is not accomplished in this work. Instead of a general system robustness measurement, we contribute here with insights and themes to explore. We do this by studying a simple system and conducting experiments with it. We include nonlinear disturbances in that system, exploring the ideas about its physical structure and its processes. To tackle the problem, we use a simple one-dimensional electrical network as the basic case study, where we analyse (1) the physical structure (a communication line in this study) and (2) the processes running on top of it (the signals traveling to communicate, and unexpected surges). Further, two- and three-dimensional examples are shown to illustrate more generalized cases.

### 1.2. Literature Review

This work goes through the analysis of a simple transmission line (a one-dimensional electrical network) connecting a source of communication to a receiver, where undesirable signals attack the line and the communication is not working properly. Depending on the strength of the alien signal, the receiver and the source can be damaged. This is typically a problem of electromagnetic interference (EMI) caused by electromagnetic emissions coming from external/internal origins [1,2]. Techniques to protect the system against these interferences are called electromagnetic compatibility methods, which are described in classical references on EMI/EMC [3–7], succinctly put in [8]. In this research, our analysis uses the transmission line model, known as the TLM method [8–12], which is a finite difference method based on space discretization that allows computational solutions based on matrix algebra techniques [13]. The TLM method was initially developed to solve one-dimensional problems, as in a transmission line propagation, and evolved to two [14] and three-dimensional models [15,16] through complex 3D cells [17], which can solve EMI/EMC spatial problems [18] and other applications, as long as computers are becoming more powerful [19–23]. We apply the TLM concept to deal with a nonlinear filter known as "varistor" [24,25], developing an algorithm to compute, through an iterative procedure, the dynamical analysis of the network under study [26]. The varistor comes to improve network communication performance. It behaves as a nonlinear filter [27–32] connected to the electrical network to respond to the alien disturbances. Based on this simple and practical electrical problem, the paper points out suggestions to a generalized dynamical network model, considering complex systems frameworks [33,34] and complex network measurements [35–37] as basic paradigms.

### 1.3. Paper Structure

The paper is structured as follows. Section 2 describes the basic concepts of electromagnetic interference, opens a short discussion on the robustness of networks, and describes a generalized network model for low voltage transmission lines. Section 3 presents an overview of linear and nonlinear filters, and then provides a description of the zinc-oxide (ZnO) variable resistor, known as a "varistor". The computational model of a ZnO varistor filter is developed to match the generalized network model of low voltage transmission lines, described in the previous section. Finally, Section 4 presents the results, a discussion about applications, and what are the perspectives for dynamical models of robustness measurements.

## 2. Electromagnetic Interference on Low Voltage Lines—Network Model

### 2.1. A short on Electromagnetic Interference

Low-voltage metallic lines in use for communication signals are very sensitive to small voltage fluctuations. Any disturbances in these signals could result in data changes, i.e., communication errors will be spread out through the system. One of the challenging factors of variances in metallic lines is the electromagnetic interference (EMI) [1,2], caused by electromagnetic emissions originated externally and/or internally. As they are, in general, not controllable, electromagnetic compatibility [1,2]

techniques should be applied to the recipient (the metallic line, in this example) to protect it from these undesired emissions.

Electromagnetic compatibility (EMC) is defined as the recipient's ability to work satisfactorily in a particular electromagnetic environment. Work satisfactorily means, firstly, it should be able to function well and secondly, it should not interfere electromagnetically in its vicinity [1–7].

Electromagnetic interference occurs when there is the involvement of a source of interference, an interfered recipient and a path coupling source and recipient (Figure 1). This structure, source/path/recipient, allows simple visualization of the interference problems and possible ways of control and compatibility, which are: (1) eliminate interference next to the source; (2) eliminate in the path; (3) or eliminate in the recipient.

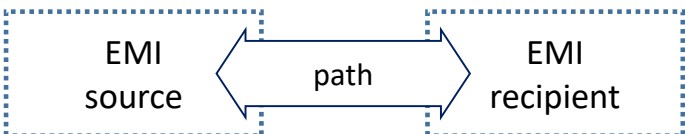

**Figure 1.** Electromagnetic interference coupling model.

1)   Eliminate EMI next to the source. If the source can be accessed, suppressing EMI could be done by, for instance, moving it away, replacement, shielding, using output filters and other suppressor devices, or by altering its internal design of circuits and components. So far, this is the best to do and the source will not be a cause of interference for the whole environment. However, in general, the source is not under control and cannot be accessed.

2)   Eliminate EMI in the path. The second possibility is doing adjustments on the transmitting path. Electromagnetic waves can be transmitted through conductive means, such as cables or "earth", (conducted EMI) or can be radiated through the environment. The alternatives to prevent the interference are the removal of the recipient equipment, or shielding it against radiated interference, and applying various types of suppressors and filters for undesired conducted disturbances.

3)   Eliminate EMI in the recipient. If it is neither possible to deal with the source nor the path, the last alternative is the elimination of EMI within the recipient. This could be a difficult task and will involve the study of the environment, the existing noise, possible transients, and surges, which will lead to a set of solutions such as physical system layout, distributed filters, suppressors and shielding.

Before we go further with the problem of interferences and the issues they can cause, let us analyse a hypothetical communication network system, the metallic line (which will be the recipient in our following case study).

*2.2. Robustness of a Simple Communication Network System*

For this work, we consider a simplified model of a communication system composed by (1) a signal source, (2) a line and (3) a receiver (this communication system, as a whole, is the EMI recipient represented in Figure 1. Let us start from this one-dimensional model to grow for a general view of dynamical networks, n-dimensional, subject to any kind of external perturbations and/or internal structural features and fails, if any.

Figure 2 (top) illustrates this simple communication system, where a squared signal is sent through a one-dimensional line from a source S to a receiver R. This signal could be, say, a +5 volts square traveling on a metallic transmission line towards the receiver R to communicate the meaning of "1", whereas a 0 volt would communicate "0". We then consider that same hypothetical line as a set of "n" nodes N interconnected by line-segments L (Figure 2, bottom), where the electrical signal travels from node to node in discrete time-steps.

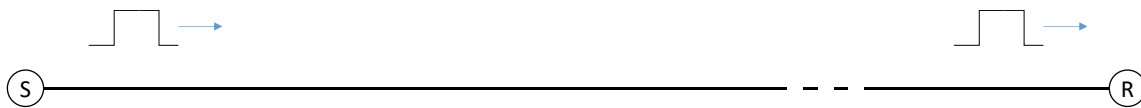

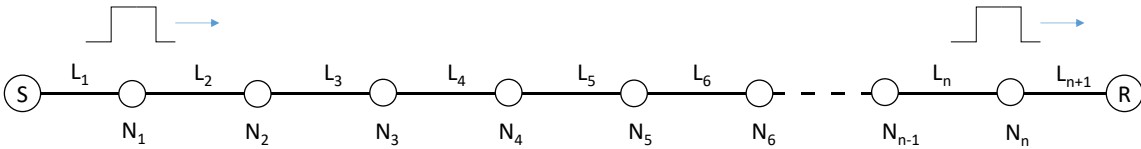

**Figure 2.** A simple one-dimensional model where a squared signal is sent through a line from a source S to a receiver R (**top**). The same line (**bottom**) considered as a set of "n" nodes N interconnected by line-segments L, where the squared signal travels from node to node in time-steps.

If we have an ideal line, the signal will travel at light speed from node to node and we can easily know the duration of each time-step according to the length of each segment. The signal will arrive at receiver R with the same shape as it was generated by S. This approach of time-steps and line-steps will afford us towards computational models to deal with a broad range of issues, as will be detailed below.

However, there are no perfect lines in real world. As pointed out before when discussing interferences, external perturbations and internal characteristics could affect the signal and its readability. Figure 3 shows illustrative examples of unpredictable interferences originated by an external unknown source. The surge voltage strikes the line at an undefined node and propagates through the line. If it happens at the time when the squared signal is passing through that very node (Figure 3, top), the signal will be distorted (compared to the original) and then reaches the receiver R with a very different shape. If the surge strikes the line before the squared signal (Figure 3, middle), it changes the original signal sequence at R, leading to a different message reading. Instead of a surge, a transient noise could hit the node, causing the electrical squared signal to become completely distorted and unrecognizable (Figure 3, bottom). These three somewhat unrealistic cases are just illustrative issues of many other possible ones, and R will read very different messages than expected by the sender S.

Despite the line´s physical structure being robust (and, in this case, ideal), the line´s functional process is not, considering that the message was distorted from the original by an external factor. Measuring the difference between the original and resultant signals shows us the damaging caused by some functional processes.

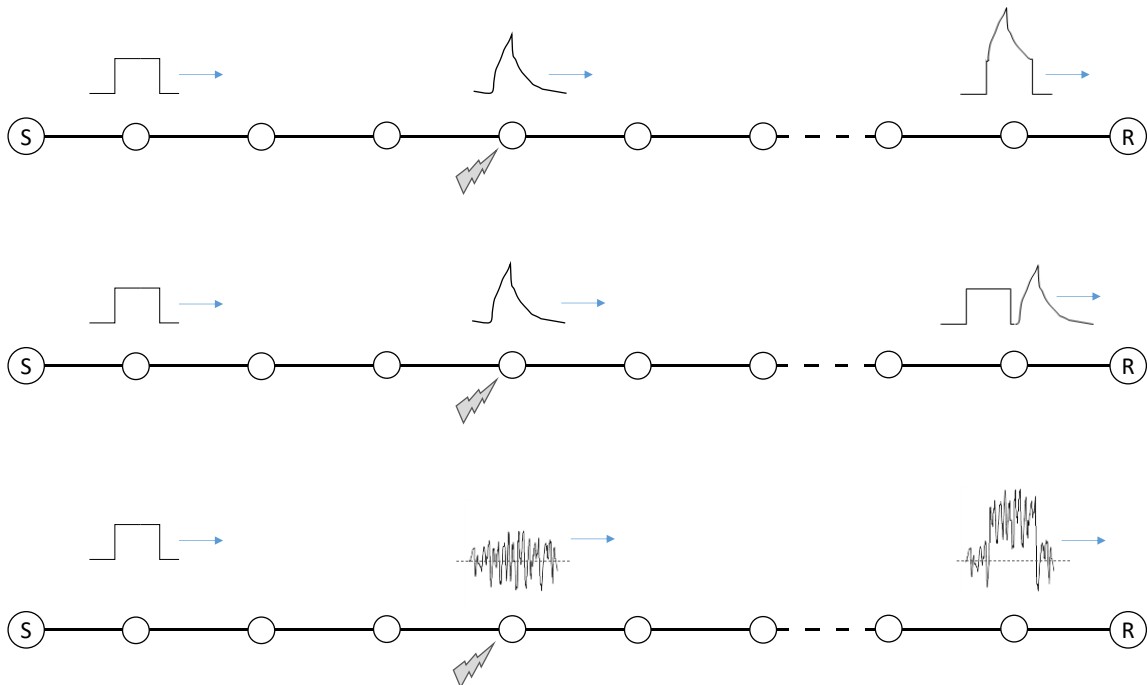

**Figure 3.** An interference originated by an external source strikes the line at a random node, propagating through the line. It may happen at the time when the squared signal is passing, distorting the signal that reaches the receiver R (**top**) or maybe dislocated in time, changing the original signal sequence at R (**middle**). If a noise hits the node (**bottom**), the squared signal could be completely distorted.

Figure 4 illustrates other cases of distortions caused by interference. For instance, a wave caused by lightning, adding a signal that changes a large sequence of bits. If the wave is too long, a whole sequence will be damaged and a very different message will be read at R. Also, other kinds of interference can multiply the original voltage by undefined factors. External interferences, their magnitude, and shapes can be unexpected and out of control. To tackle these problems, we could apply elements that modify the line *structural* features, aiming to give better responses on the line *functional* features. These two levels, structural features and functional features, should be helpful to generalize a measure of robustness for networks.

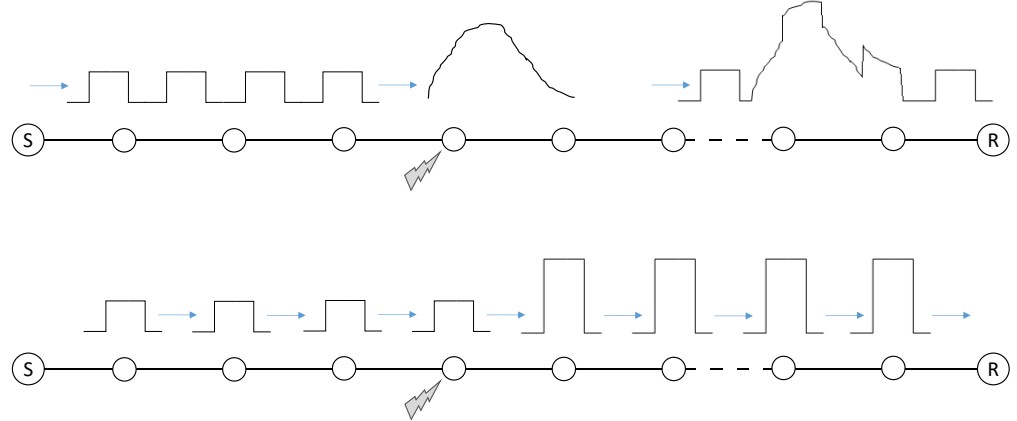

**Figure 4.** Other examples of interference, such as a wave caused by lightning, adding a signal that changes a large sequence of bits (**top**), or multiplying the voltage of the original square by an undefined factor (**bottom**). External interferences, their magnitude, and shapes can be unexpected and out of control.

If external interferences can affect the functional features level, internal structure features can affect it as well. Two examples in Figure 5 illustrate how internal structure features can damage or destroy a message—that is, how the functional features will be affected by the structure and at what amount. Line structural variations could modify the signal sent by S in such a way that R could misunderstand it, as shown by the first example (Figure 5 top). The electrical line may present a different and increasing capacitance from node N5 and beyond, significantly distorting the signal along its length, and the traveller square voltage signal is reshaped (by obvious illustrational purposes the figure exaggerates the representation). The second case shows a failure that intermittently interrupts the line-segment between N5 and N6, and no signal achieves the receiver R when the N5-N6 link is off (Figure 5 bottom). Figure 5 shows that physical structure features may have varying degrees of robustness, i.e., from relatively distorting the system performance (the functional processes still work) to completely outages (no processes occurring at all).

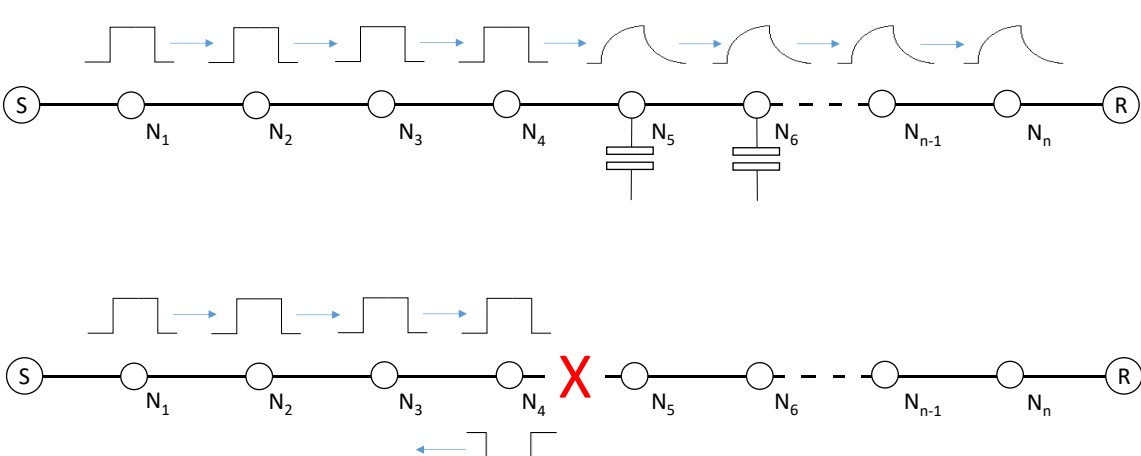

**Figure 5.** Line structural variations could modify the signal sent by S in such a way that R could misunderstand it (**top**) or even do not receive it (**bottom**). These inherent characteristics affect the communication of internal factors. A metallic line has a different and increasing capacitance from node N5 and beyond, significantly distorting the signal (top example), or a failure that intermittently interrupts the line-segment between N5 and N6 (bottom example).

In short, the robustness of the system illustrated in Figures 3–5 is a result of two combined factors, which we call **structural** and **functional**. The structure depends on the features related to the system physical structure, and the functional factor has to be with the processes running on that structure. Intrinsically interrelated, the two factors could define degrees of the system robustness. Defining a degree of system robustness is a subject that escapes from the purpose of this paper and will not be addressed in this work. Instead, a model to dynamically deal with the system is developed later, showing ways to enhance the low voltage line robustness and sustainability.

### 2.3. Low Voltage Lines—Network Model

Going further with the simple telecommunication line above described, let us see it as a metallic transmission line as illustrated in Figure 6, where a transmission line connects source S to receiver R and is split into n nodes N (Figure 6, top). The transmission line parameters are its inductance and capacitance, and losses are determined by its resistance and conductance. Figure 6 (middle) shows a representation of the line segments and their distributed inductances and capacitances (for design clarity the resistance and conductance are not shown). These parameters are specific structural characteristics.

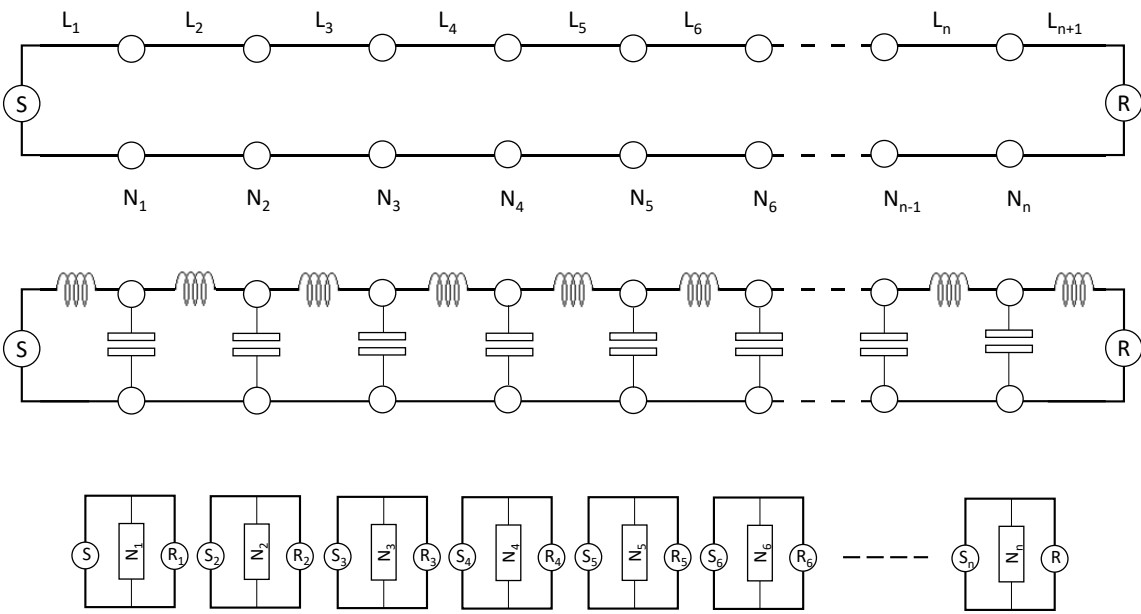

**Figure 6.** Metallic transmission line connecting source S to receiver R, considering *n* nodes *N* (**top**). The transmission line segments have distributed inductances and capacitances, their specific structural characteristics (**middle**). Each line segment can be seen as a transmission line itself, considering also distributed sources $S_n$ and receivers $R_n$ (**bottom**). This line segment is called a "node" of the network.

Back to the approach of time-steps and line-steps, each node N and its boundary line segment is modelled as a transmission line itself, considering distributed sources $S_n$ and receivers $Rn$ (Figure 6, bottom). This approach is called the transmission line method (TLM) and is well established [8–12], standing as a finite difference method in the time domain. The finite difference methods are based on space discretization (grids) that allows computational solutions based on matrix algebra techniques [13].

Each node $N_n$ is seen as a transmission line itself connected to the previous ($N_{n-1}$) and the next transmission line ($N_{n+1}$), from the source to the receiver. The signal moves from node to node in time-steps—that is, from one transmission line segment to the following transmission line segment. The basics of the whole model, in one, two and three-dimensional networks, were described in [9,11], whereas a different three-dimensional cell was presented by [12].

Our approach considers the example showed in Figure 7, which is derived from [8,12]. The top picture shows segments *x* and *x* + Δ*x* of a transmission line with parameters *L, C, G* and *R* (inductance, capacitance, conductance, and resistance respectively). The node $N_n$ comprises these parameters around the point *x*, which is modelled as an impedance $Z_0$ with losses represented by *R* and *G* (respectively resistance and conductance for the segment with length Δ*x* (Figure 7 bottom). This representation is called the transmission line method model, where the node $N_n$ is a transmission line itself, with the incident and reflected voltages for each time-step. A complete description of the one-dimensional method and its equations were given in [8,12]. Below, we summarize its most important concepts and equations, which will be useful to model the system's dynamical behaviour and the nonlinear filter in the sections following.

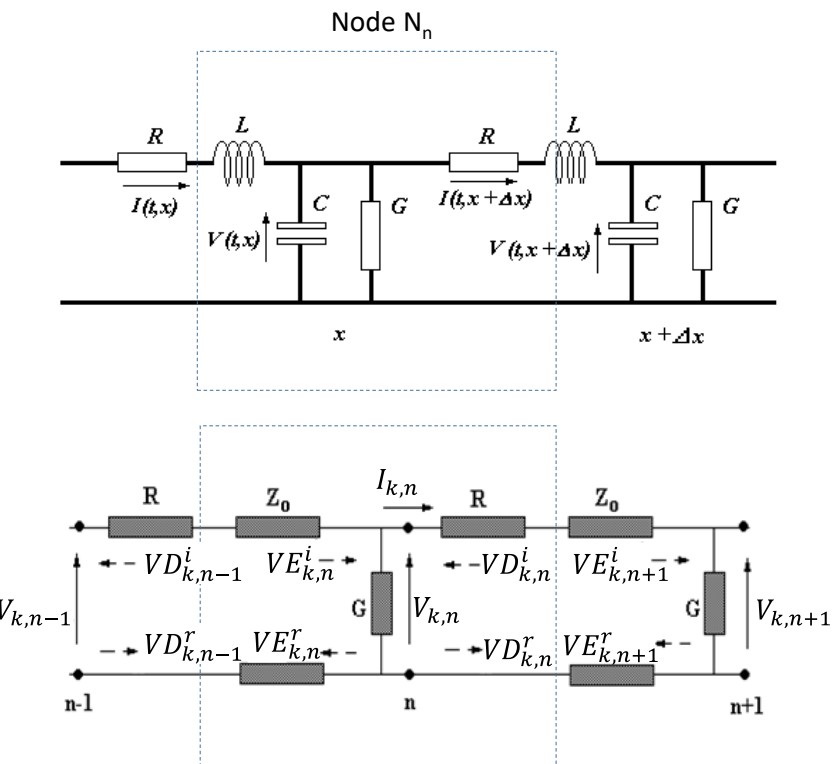

**Figure 7.** Segments of a transmission line $(x, x + \Delta x)$ with distributed parameters $L, C, G$ and $R$ (top). The same segments represented as transmission line method models, where the node "n" is a transmission line itself, with incident $V^i$ and reflected $V^r$ voltages for each time-step (bottom).

For each node, the characteristic impedance $Z_0$ is given by

$$Z_0 = \sqrt{\frac{L_d}{C_d}} = \sqrt{\frac{L}{C}} \tag{1}$$

where $L_d$ and $C_d$ are the distributed line inductance and capacitance, respectively. The wave speed propagation $v$, for this line, which is also the speed propagation in each line segment, is

$$v = \frac{\Delta x}{\Delta t} = \frac{1}{\sqrt{L_d C_d}} \tag{2}$$

The wave travels from one node to the next with a time interval (time-step propagation) defined by

$$\Delta t = \sqrt{LC} \tag{3}$$

Each node is an independent line, which is interconnected to adjacent lines (see Figure 6). The connection between these lines is made according to the incidence and the reflection of electrical waves. Calculating iteratively, for all nodes, the incident and reflected voltages and their resultants, we can know at each point of a line, at its source and its receiver (load), at any time, the voltage and current levels.

The line segment, or node, pictured in Figure 6 (bottom) is modelled as shown in Figure 8. Each line segment is characterized as a transmission line with a source $S_n$ and a receiver $R_n$ (we call it node $N_n$). The Thévenin equivalent for this node $N_n$ shows voltage sources $VE_{k,n}$, and $VD_{k,n}$ and line parameters $G$, $R$ and $Z_0$, where $k$ is the iteration and $n$ is the node (Figure 8, right). Losses caused by line resistance and conductance are considered and calculated within the node. This is an important feature of this method when we consider the node as an element of a dynamical network. It means that

the network behaviour, as signals travel between nodes through existent connections, will be modified by the specific characteristics of each node, and not only by the general structure of the network (in other words, it is not a view of a graph in its static mode).

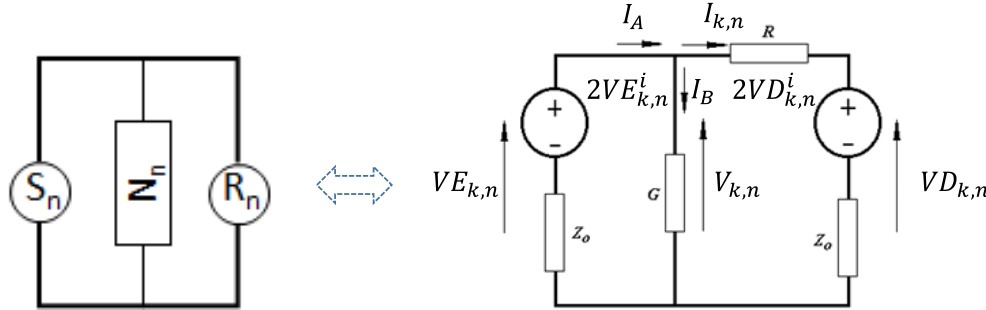

**Figure 8.** Each line segment (the node $N_n$) is a transmission line with a source $S_n$ and a receiver $R_n$ (**left**). Its electrical model - Thévenin equivalent (**right**) consists of voltage sources $VE_{k,n}$ and $VD_{k,n}$ and line parameters $G$, $R$ and $Z_0$, where $k$ is the iteration.

The voltage and current equations for this generic node $N_n$, detailed in [8], are:

$$V_{k,n} = VE_{k,n} \tag{4}$$

$$VD_{k,n} = 2VD^i_{k,n} + I_{k,n}Z_0 \tag{5}$$

$$I_{k,n} = \frac{V_{k,n} - 2VD^i_{k,n}}{(R + Z_0)} \tag{6}$$

$$V_{k,n} = \frac{\frac{2VE^i_{k,n}}{Z_0} + \frac{2VD^i_{k,n}}{R+Z_0}}{\frac{1}{Z_0} + \frac{1}{R+Z_0} + G} \tag{7}$$

From Figure 7 (bottom), we know that the relationship between incident ($V^i$) and reflected ($V^r$) voltages at iteration $k$ for the node $n$ can be obtained by adding their contributions, which determines its total voltage on the left ($VE_{k,n}$) and right ($VD_{k,n}$), as seen in the following equations:

$$VE_{k,n} = VE^i_{k,n} + VE^r_{k,n} \tag{8}$$

$$VD_{k,n} = VD^i_{k,n} + VD^r_{k,n} \tag{9}$$

Reflected voltages from within the node $N_n$ will be the incident voltages to the adjacent nodes $N_{n-1}$ and $N_{n+1}$ on the next time step $k+1$, as follows:

$$VE^i_{k+1,n} = VD^r_{k,n-1} \tag{10}$$

$$VD^i_{k+1,n} = VE^r_{k,n+1} \tag{11}$$

This allows the computation of the wave propagation (from time-step $k$ to $k+1$), and any losses during the propagation will be considered internally at each node when calculating $V_{k,n}$ (Equation (7)). The source $S$ is considered in the first node, whilst the receiver $R$ in the last one. The detailed explanation and development of equations can be found in [9] and [12].

Inductance and capacitance are natural filters that affect low and high frequencies, respectively, for the line described. Figure 5 (top) shows, for instance, how a high capacitance could distort the original signal. If we apply external filters to this line, like additional capacitances, we can easily see attenuation for high frequencies wherever we need them along the line. However, if we have

unexpected interferences, as explained above, an alternative is the use of special nonlinear filters to deal with these issues, but not affecting the usual line operation. In other words, these nonlinear filters will work as structural transitory features. If the functional features are working properly, no alterations will be seen at the structural level. If a threshold is achieved and the functional level could be out of control, the filter will temporarily come to be part of the structure, operating to manage minimum functional system performance. The system's robustness depends on the interaction between the structure features (the line physical characteristics) and the functional features (the expected process characteristics). For this case study, the nonlinear filter is the element which varies its action, aiming to maintain the network adequate operation (i.e., the network robustness).

Connecting this kind of structural transitory filter to one or more nodes of the network will add robustness to the whole system (Figure 9). To analyse the behaviour of the nonlinear filter, its dynamical model must be developed and be implemented within a node. The following sections describe the physical characteristics of a semiconductor nonlinear filter and how it is applied to our generic model.

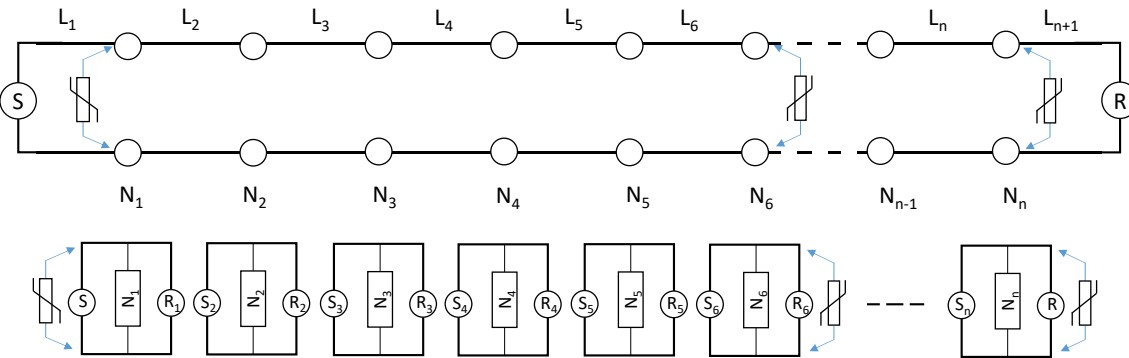

**Figure 9.** A nonlinear filter applied to one or more points of the metallic line (**top**) and its representation as applied to the respective network node (**bottom**). The filter will be temporarily part of the structure according to unexpected functional feature variances (for example, alien attacks).

## 3. Linear and Nonlinear Filters

### 3.1. Overview

In the example above, we see that capacitance and inductance can alter the wave time speed (Equation (2)), and capacitance will filter high frequencies (example in Figure 5). This capacitance filtering will vary linearly with the wave frequency. That is, the linear filter has an output that is a linear function of its input. On the other hand, a nonlinear filter has an output that is not a linear function of its input.

In our model, the capacitances and inductances, as well the losses, are described within the node model. Any kind of disturbances or interferences on the line will hit the Receiver R. What kind of filters could be applied to the system that will not affect the normal signals, but only when these interferences appear? Nonlinear filters are suitable for this purpose because they will not be active when the signals are those normal ones expected but will be activated when unexpected and abnormal ones are surging. An appropriate model for this kind of nonlinear filter must be defined and introduced to the iterative process, giving to the network a dynamical behaviour, according to each level of abnormality.

For metallic transmission lines, the semiconductors are considered powerful nonlinear filters, and for high levels of voltage and/or energy, the metal-oxide variable resistors, or varistors, are among the most useful. Their physical characteristics and parameters, as well as their typical curves, are described below and will enable us to build their dynamic models. Although this is a very specific electrical component, the model is generic and can be applied to any other nonlinear elements, in any other dynamical systems that use this network approach.

### 3.2. Metal-Oxide Varistor

Semiconductors are very sensitive to overvoltage transients and noise in the electronic circuits, and the majority of sources of these transients are the lines attached to the electrical feeding. These lines, through the general and several times unpredictable routes they follow until the power input of the devices, are very susceptible to electromagnetic interference, despite all the actions taken to protect them.

Filters to suppress these transients and noise, even low-energy ones, must be extremely quick in responding to the rise of voltage and current, absorbing high rates of energy and maintaining the continuity of the regular electronic process during, and after, the occurrence. Components that apply to these exigencies are the variable resistors, called varistors (variable-resistor), which are composed of metal oxides.

Varistors are transient voltage surge suppressors, or TVSSs, whose resistance decreases with increasing voltage [24], i.e., varistors are components that depend on the voltage on their terminals, with a symmetrical V/I (voltage/current) characteristic curve as exemplified at Figure 10. Considering a circuit or device to protect against transients, the varistor is connected in parallel to the circuit, building a shunt that will divert the voltage excess. The value of the voltage to be diverted should be defined accordingly to the device/circuit normal operating voltage. In other words, the suppressor will deviate part of the energy (voltage and current), allowing just the necessary voltage to feed the device.

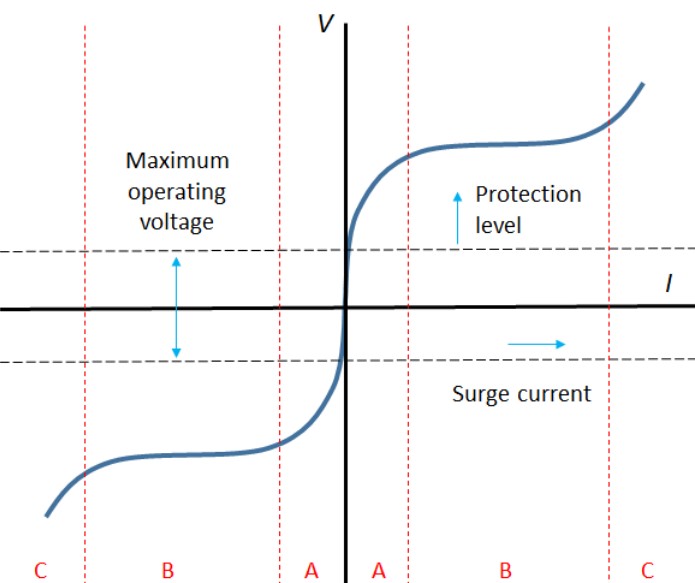

**Figure 10.** Typical voltage/current (V/I) symmetrical curve of metal-oxide varistors (not dependent on the polarity of the voltage applied, linear scale).

As the curve in Figure 10 illustrates, the resistance is very high for the range of defined maximum operating voltages (behaving like an open circuit), but after a threshold, the resistance tends to decrease with a nonlinear behaviour, which is called "protection level".

At region A, the varistor behaves almost like an open circuit ($R > 1$ Mohm), and at region C as a resistor with a low resistance value ($R\sim1$ ohm) [24,25]. At region B, its behaviour is characterized by the equation

$$I = \beta.V^\alpha \tag{12}$$

where $\alpha$ denotes the nonlinear varistor behaviour, and $\beta$ is the ceramic constant that depends on the varistor type and manufacturing. The Equation (12) is a power-law where the nonlinear factor $\alpha$ typically varies from ~10 to as high as ~30 for varistors, which have response times of <25 ns [24,25].

A typical varistor curve of a commercial product is illustrated in Figure 11, where, for simplicity, only the positive quadrant is shown in a log-log plot. Both $\alpha$ and $\beta$ can be calculated from the technical specifications of each commercial varistor, considering the values of voltage and current for the points *a* and *b* (Figure 11) and using the following equations.

$$\alpha = \frac{\log\left(\frac{I_b}{I_a}\right)}{\log\left(\frac{V_b}{V_a}\right)} \tag{13}$$

$$\beta = \frac{I_b}{V_b{}^{\alpha}} \tag{14}$$

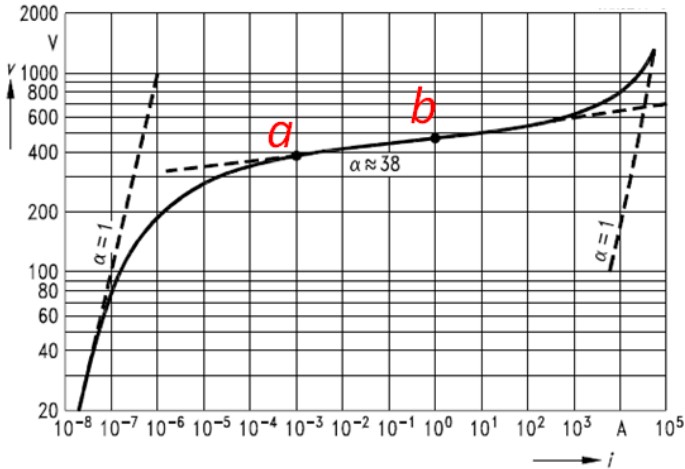

**Figure 11.** Typical V/I curve of a commercial metal-oxide varistor [9].

It is important to consider that in computational simulations the values of $\alpha$ and $\beta$ could slightly vary, depending on the points *a* and *b* considered in the varistor curve, and the manufacturer specifications. The value of $\beta$, calculated in Equation (14), should use $V$ and $I$ closer to *b* at Figure 11, as shown by experimental results (see Section 4). One could argue that $\beta$ is not really a "constant". It is true. It is approximately constant in region B (Figure 10). Anyway, the value obtained will be enough for simulations, as we will show.

Varistors are polycrystalline ceramics made by mixing metal oxides and additives. The most common varistor is done with zinc oxide, ZnO, which offers a resistance highly dependent on voltage, besides great energy capacity. The grains of zinc oxide are conductive, but the other additives are highly resistant. Figure 12 shows schematically the composition of a varistor where the grains of ZnO produce several "microvaristors", similar to Zener diodes [24]. Its physical dimension determines the electrical properties of a varistor because, as Figure 12 shows, microvaristors are connected both in series and in parallel. Therefore, the production of a varistor can determine the value of protection in terms of voltage, current, and energy. This is due to the thickness of the element (twice the thickness, twice the operational voltage due to the microvaristors in series), its area (twice the area, twice the operation current due to the microvaristors in parallel), and its volume (twice the volume, almost twice the energy absorption because there are as many twice the number of grains). Also, it is important to consider that the ceramic is compressed between metallic disks and their leads of tinned copper, adding an important capacitance characteristic to the suppressor. This high capacitance smooths the steep surge voltage edges, improving the protection level [24].

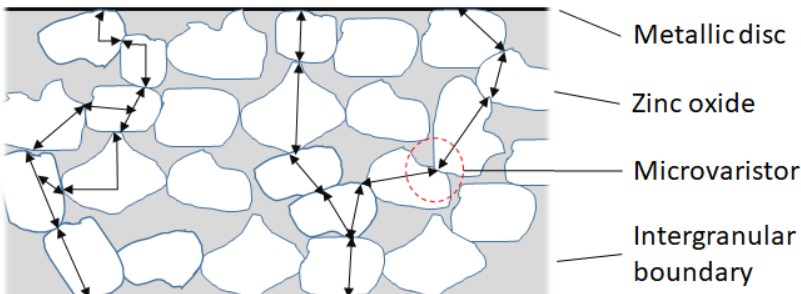

**Figure 12.** Section of a varistor showing the ZnO grains and the composition of "microvaristors" encapsulated between metallic discs.

### 3.3. The Dynamic Model of A ZnO Varistor Filter

The question that arises about the nonlinear filter is: where and how to insert it in the finite-difference time-domain TLM model above described? We saw that all parameters are calculated within the nodes at each time-step, and the dynamic occurs when reflected signals are considered the new incident signals on the next time-step, forwarding the process on. For this reason, the model of any component must be embedded in the node where it is activated. That is to say that the nonlinear filter will be computed within the node where it is connected. This procedure agrees with the concept of bringing all "decision-making" about the signal processing to within the node in this model.

However, its nonlinearity brings a novel problem for the modelling, which occurs just when the time-steps are going forward, and the nonlinearity results are considered and calculated <u>within</u> one time-step, and not between two consecutive time-steps. As we see in [12], computing a nonlinear element directly into a node can lead to unwanted oscillations. This happens because the varistor response at a given time step may give rise to a large voltage variation for the next iteration (within the node where it is attached), which in turn does not allow an adequate voltage to be found at the next calculation, causing the process not to converge as expected. The solution to this problem is to solve the nonlinear equation within the node, but only during the connection between two consecutive time-steps (e.g., $k$ and $k + 1$) [12].

The nonlinear component model considers the component (varistor) divided into two parts: firstly, a capacitor of linear behaviour (detailed at [8]) and, secondly, a resistor with a nonlinear equation to be solved by a specific calculation process. The capacitor is modelled as a stub, to where the voltage is reflected at each iteration ($k$) and where a new incident voltage returns to the circuit in the next iteration ($k + 1$). In the stub model, the capacitor is introduced as a shunt. Thus, the signal does not cross it continuously. Instead, it is forced to shift inwards and then back to the line, using a total time $\Delta t$ to complete this process. The capacitor extremities are represented as an open circuit, which the wave reaches at time $\Delta t/2$, reflecting and then returning to the line at another $\Delta t/2$, with a reflection coefficient equals +1 [8]. The resistor nonlinear behaviour calculation is done in this range, which implies that the new incident voltage on the node is different from the previously reflected voltage, i.e., $Vvar^i_{k+1}$ is different from $Vvar^r_k$, differently to what happens for a linear capacitor filter [8,12]. Then, taking into account the reflected and incident voltage differences within the nonlinear component, the current $Ivar$ circulating through the varistor will be given by the equation [12].

$$Ivar = \frac{Vvar^r_k - Vvar^i_{k+1}}{Zvar} \tag{15}$$

where Zvar is the varistor characteristic impedance (which considers its capacitance), $Vvar^r_k$ is the reflected voltage to the varistor at time k, and $Vvar^i_{k+1}$ is the incident voltage coming from the varistor to the node where it is connected at moment $k + 1$ [12]. The voltage in the varistor is given by

$$Vvar = Vvar^r_k + Vvar^i_{k+1} \tag{16}$$

By substituting the expressions (15) and (16) in Equation (12), we can finally get

$$\frac{Vvar_k^r - Vvar_{k+1}^i}{Zvar} = \beta\left(Vvar_k^r + Vvar_{k+1}^i\right)^\alpha \tag{17}$$

The resolution of Equation (15) must be done at each iteration to provide the new incident voltage $Vvar_{k+1}^i$, thus allowing the connection and continuity of the calculation process. Due to the nonlinear and non-quadratic condition of Equation (15), a specific method of calculation was developed and described in [12] and implemented in [26]. When the calculation reaches the admitted error value considered in the method, the value of the new incident voltage is obtained and the connection will be given at the next moment. The values of $\beta$ and $\alpha$ are obtained from the manufacturer's technical specifications (which may present distortions accordingly to each manufacturer or fabrication method) [24,25]. Other studies and applications were given in [27–31].

## 4. Results, Discussion and Perspectives

### 4.1. Results of Practical Experiments and Simulations

The experiments prepared to test our model considered a communication metallic line such as the one pictured in Figure 9, with the nonlinear varistor filter applied on the last node, close to the receiver R.

Table 1 shows the distributed line parameters (resistance, capacitance, and inductance per metre), and Table 2 shows the characteristics for the zinc-oxide varistors S05K11, S10K11 and S10K14 [24] used in the tests. Their constants $\alpha$ and $\beta$ were determined by voltages and currents at points $a$ and $b$ according to technical plots from [24], values are shown in Table 1 (see also Figure 11). Simulations were done with the computer Python code available at [26].

**Table 1.** Distributed line parameters (resistance, capacitance and inductance per metre).

| Line Parameters (Distributed) | | |
|:---:|:---:|:---:|
| R (Ω/m) | C (F/m) | L (H/m) |
| $1.0 \times 10^{-5}$ | $1.0 \times 10^{-10}$ | $2.5 \times 10^{-7}$ |

**Table 2.** Characteristics for the zinc-oxide varistors used in the tests.

| Varistor Model [tdk] | Metal Oxide | Disk Diammeter (mm) | C (nF) | Nominal Voltage (V) | Assumed Values According to Technical Specifications | | | | | |
|:---:|:---:|:---:|:---:|:---:|:---:|:---:|:---:|:---:|:---:|:---:|
| | | | | | $V_a$ (V) | $I_a$ (A) | $V_b$ (V) | $I_b$ (A) | $\alpha$ | $\beta$ |
| S05K11 | ZnO | 5 | 1.6 | 11 | 23 | 0.01 | 34 | 1 | 11.78 | 9.04 |
| S10K11 | ZnO | 10 | 6.8 | 11 | 23 | 0.01 | 30 | 0.5 | 14.72 | $8.93 \times 10^{-23}$ |
| S10K14 | ZnO | 10 | 5.2 | 14 | 28 | 0.01 | 36 | 0.5 | 15.56 | $2.97 \times 10^{-25}$ |

Using the same line parameters of Table 2, with a varistor S05K11 protecting the receiver load, a practical experiment was done in the laboratory. A 50 V squared wave, 6 μs, was applied to the line as an alien attack and the result is shown in Figure 13, where clamping values after filtering vary around 31 volts (compare to clamping simulations shown in Appendix A, Figure A1, top-left).

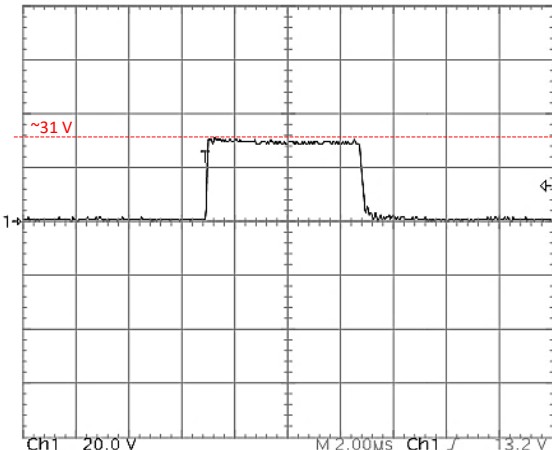

**Figure 13.** Laboratory test for a 50V squared wave, 6e-6 sec, applied on a load protected by varistor S05K11 [24], results collected using the Tektronix oscilloscope [32]. Due to ceramic manufacturing and assembly peculiarities, results may vary slightly from component to component.

Several simulations of our dynamical nonlinear model using zinc-oxide varistors as examples of application are shown at Annexes A, B and C [26]. Annex A shows the results for the ZnO varistor model S05K11 [24]. This commercial product is composed of a metallic disk of 5 mm diameter comprising the zinc-oxide ceramic, nominal voltage 11 V and capacitance 1.6 nF. The left side pictures at Annex A show squared pulses of 6 µs (50, 75, 100 and 150 V high) achieving the load and the resultant voltages after the nonlinear filter clamping. The right side shows atmospheric surges and correspondent clampings. The atmospheric surge is a standard definition to represent typical lightning surges, characterized as an impulse with the format 1.2 × 50 µs, which means that the surge rises from 10% to 90% of the maximum voltage in 1.2 µs, decreasing to 50% in 50 µs [9]. This curve format is coded to simulate the lightning electromagnetic interference at [26]. The top-left plot is the simulated computational correspondent to the practical result shown in Figure 13. Comparing pictures from the left to the right, one can realize the consistency of the clamping values for both surge types and how, for the atmospheric surge example, it smoothly decreases as the curve goes down.

Annex B shows the results for the ZnO varistor model S10K11 [24]. This product is built by metallic disks of 10 mm diameter containing the zinc-oxide, nominal voltage 11 V and capacitance 6.8 nF. The left side shows squared surge pulses and the right side shows atmospheric surges and correspondent clampings. When compared to the previous varistor S05K11, which has the same nominal voltage, it is easy to notice how the capacitance influence on the surge steep rise and fall filtering provokes a smooth response. This response is quite interesting to prevent damage on subsequent loads. This additional filtering effect is obtained by the higher capacitance, which is a very simple constructive improvement by just doubling the disk diameter. Additionally, as the diameter is bigger, also its energy absorption is higher, which implies a large number of industrial applications.

Considering now the same varistor diameter (10 mm), but a higher nominal voltage, Annex C shows the results for the ZnO varistor model S10K14 [24]. This one has capacitance 5.2 nF, and nominal voltage 14 V. The left side pictures show squared surge pulses and the right side shows atmospheric surges and correspondent clampings. Retaining the same benefits of the previous varistor, that is, capacitance filtering and higher energy absorption, this product has a nominal voltage slightly higher (14 V) that results in a higher voltage clamping. In the atmospheric surge, clamping is noticeable in the transient passage from a nonlinear process to the return to a normal curve. It happens in the region "A" of the symmetrical curve of the metal-oxide varistor (see Figure 10—Typical V/I symmetrical curve of metal-oxide varistors).

### 4.2. Discussions, Perspectives and Conclusions

Nonlinear filter behaviour in the real-world is difficult to predict and we can only assume a range of possible responses. However, a simulation model can have better results if we compute the signals dynamically and therefore we have a narrow range of results to work with.

Looking to the varistor plots in Appendix A, Appendix B, and Appendix C, clamping zones vary according to the physical structure of each component, and with the processes running on top of these structures. The complete communication system, i.e., line, sender, receiver, filter, and interferences, create a complex adaptive system.

Comparing the three different varistors used in our simulations, one can realize the slight result differences when each filter is operating, even when they have the same nominal voltage clamping. Additional features, as the inherent capacitance, can affect the outcome (in these cases, positively if we consider they can smooth the surges). Another interesting result happens when lightning interference strikes the system (not unusual at all). In this case, the curve after the varistor filter slowly varies, decaying as the strike decays, until the point where the nonlinear filter suddenly becomes linear again (see the plots at the right side at Figures A1–A3). This happens to agree with the transition from the varistor nonlinear region to the linear region, as shown in Figure 10, region B.

The case study presented is a one-dimensional model and does not show signals in the surroundings of the line. If the line is an antenna, for example, how can we predict the near- and far- electric and magnetic fields? The one-dimensional model can be extended to two-dimensional or three-dimensional models, applying the same process of reflected and incident signals travelling through the connections. Figure 14 illustrates these models considering Cartesian coordinates, which are useful for electric wave propagation—several models were developed to solve these problems and presented in [8,9,12]. If the problem is not Cartesian coordinate dependent, the dimension of the resultant matrix will consider each node degree. Each reflection will be the result of a function, dependent on the node characteristics. This process was used when we modelled the varistor, considering its internal nonlinear characteristics to give a response for each incident signal.

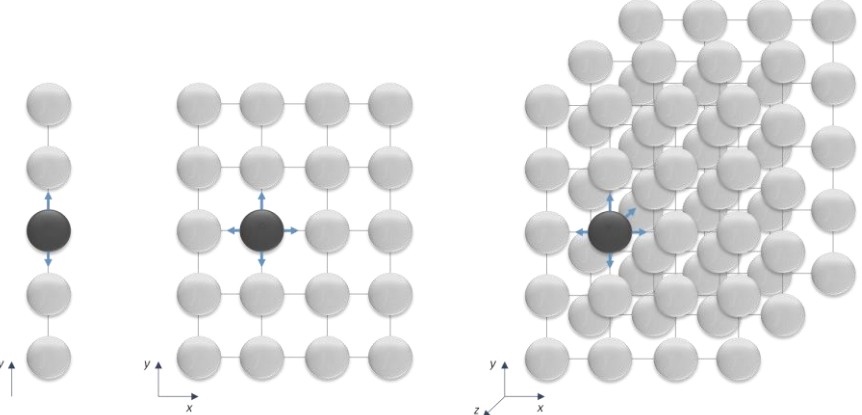

**Figure 14.** One, two and three-dimensional grids, where an excited node (input) sends signals to its neighbours, starting an iterative time-step incident/reflection process. These examples show static network structures, with dynamical processes running on top of them.

The nonlinear filter was applied to the line to guarantee communication sustainability (the communication robustness, in other words). The varistor adds to a specific node a transitory structural characteristic: if a threshold of instability is achieved, i.e., there is an unacceptable variation in the process, the internal parameters of the node changes to return to a "normal" condition, correcting the instability. The acceptable condition, for this case study, means a range of typical signals recognizable by both sender and receiver. It is interesting to notice that the nonlinear component transitorily changes the internal structure of a node, changing its features to provide sustainability for the process. Here we

see the two factors mentioned before: (1) the physical structure and (2) the processes running on top of it. Figure 15 illustrates these two components, structure and processes. The processes corresponding to the dynamic component, and the structure is the static counterview. Strictly, the structure is not static but is considered like this in the model when we freeze one moment, *t*, and run the processes on top of it to connect *t* to *t* + 1. At *t* + 1 we could, or not, have a new structure, and again the process re-starts.

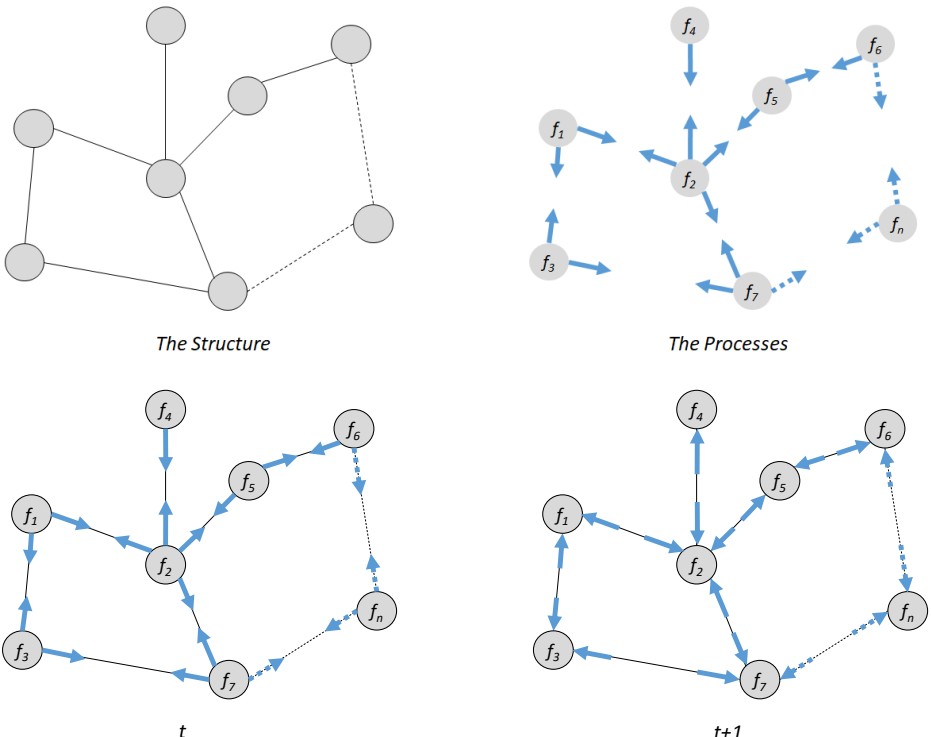

**Figure 15.** The system as a static network structure with dynamical processes running on top of it. The dynamical approach considers reflected signals (time-step *t*) and incident signals (time-step *t* + 1), dependent on the connections and node functions $f_n$.

The idea in this work is that system robustness is simultaneously dependent on the structure robustness and the process robustness. Studies on system robustness are present in theories about complex systems [33,34], along with several methods and measurements describing complex network [35], graph and dynamic communities [36], social network analysis [37], just to cite a few. However, these measurements are mainly dealing with the network "structure", not the processes. Here we argue that both, in our model, must be considered within the iterative computation, bringing each node and its connection parameters together to calculate the system dynamical evolution. The nonlinear application presented is an example of this approach. Our future research will focus on a model to measure the system sustainability based on the *structure* and the *processes* framework, as outlined in Figure 15.

**Funding:** This research received no external funding.

**Acknowledgments:** The author acknowledges Karoline Wiesner for her contributions with discussions, ideas and several suggestions for the final version of this paper, and The University of Bristol and Unisul Universidade do Sul de Santa Catarina for their administrative support.

**Conflicts of Interest:** The authors declare no conflict of interest.

## Appendix A

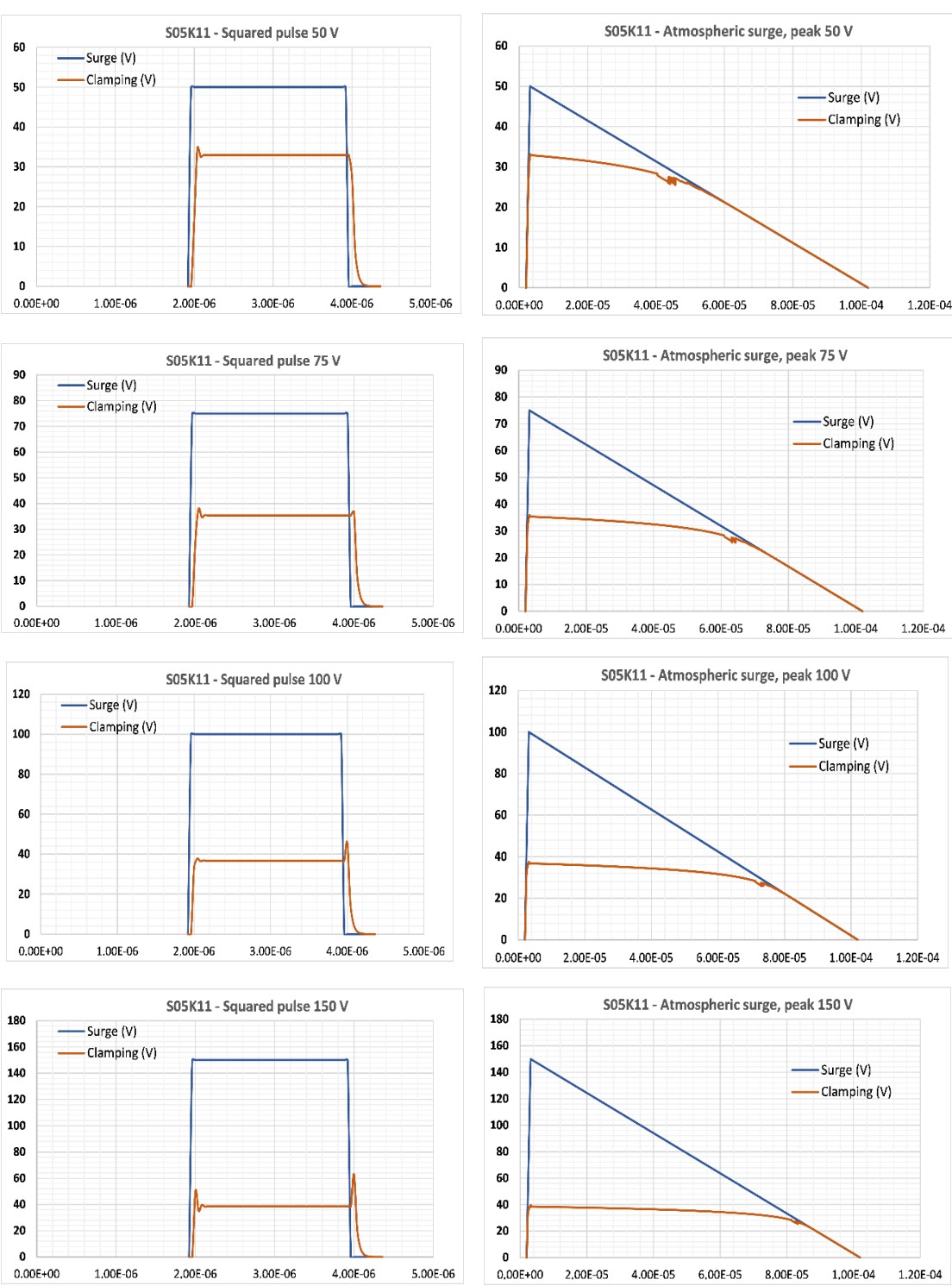

**Figure A1.** Results for the ZnO varistor model S05K11 [24], which is composed with a metallic disk of 5 mm, nominal voltage 11 V and capacitance 1.6 nF. The left side pictures show squared pulses achieving the load (blue line, 50–150 V) and the resultant voltages after the nonlinear filter clamping. The right side shows atmospheric surges and correspondent clampings. Note the consistency of the clamping values for both surge types and how it smoothly decreases as the surge goes down for the atmospheric example.

## Appendix B

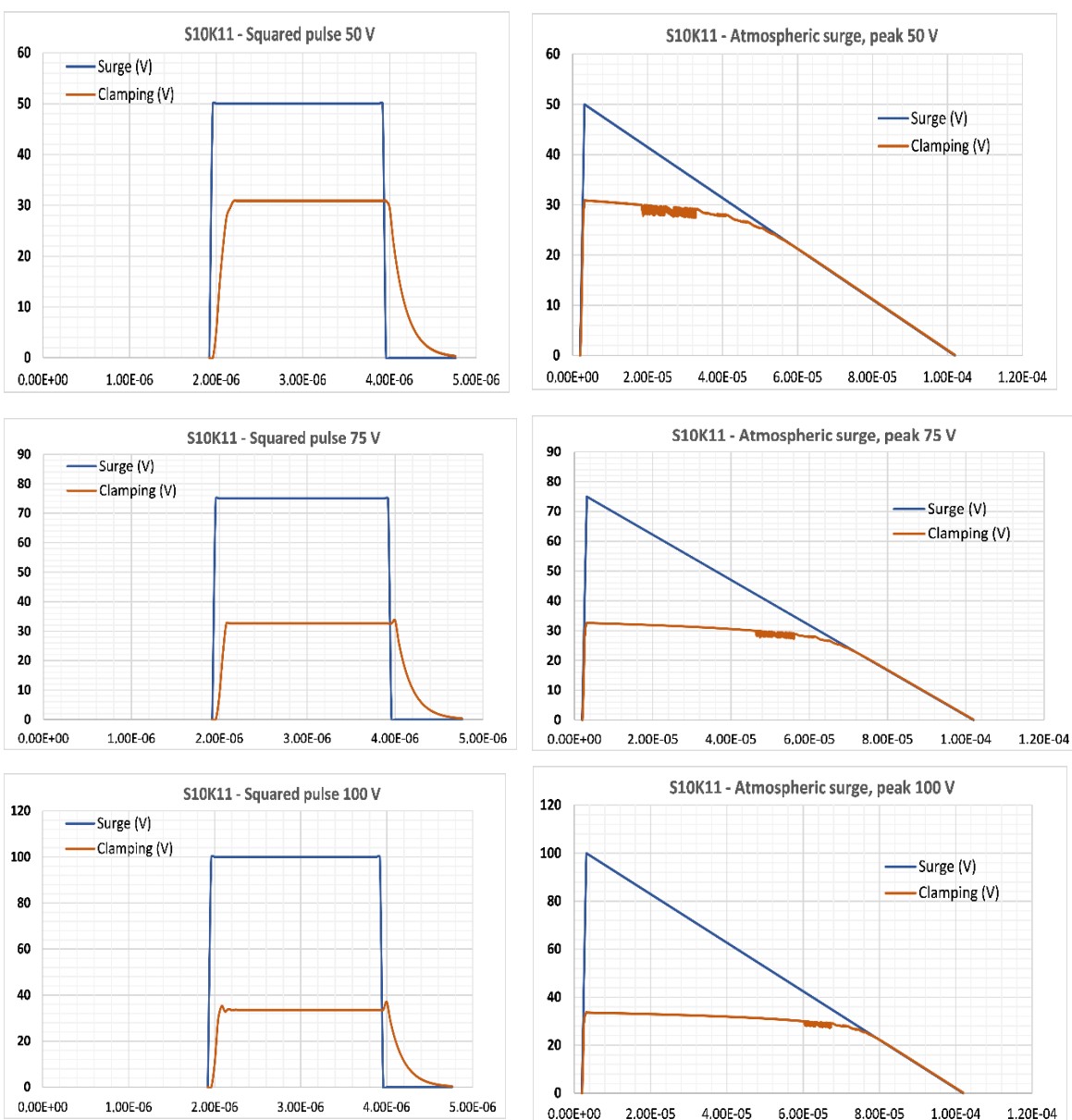

**Figure A2.** Results for the ZnO varistor model S10K11 [24], which is composed by a metallic disk of 10 mm, nominal voltage 11 V and capacitance 6.8 nF. The left side pictures show squared surge pulses and the right side shows atmospheric surges and correspondent clampings. Note the capacitance influence on the surge steep rise and fall, which is quite interesting to prevent damages on subsequent loads. This additional filtering effect is obtained by the higher capacitance, which is a very simple constructive improvement compared to the previous S05K11. As the diameter is bigger, also its energy absorption is better.

## Appendix C

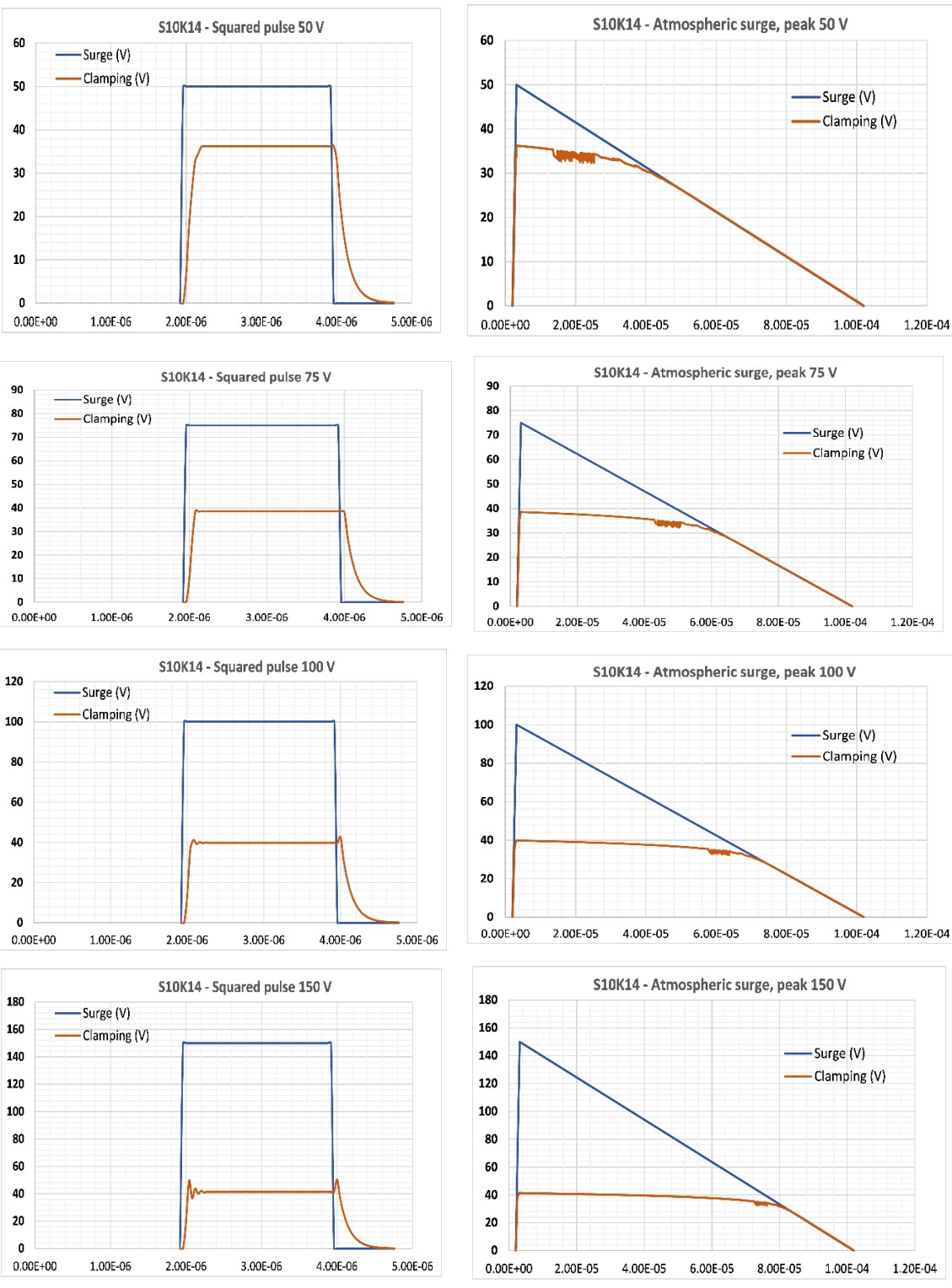

**Figure A3.** Results for the ZnO varistor model S10K14 [24], which is composed by a metallic disk of 10 mm, nominal voltage 14 V and capacitance 5.2 nF. The left side pictures show squared surge pulses and the right side shows atmospheric surges and correspondent clampings. Note here again the capacitance influence on the surge steep rise and fall. However, its nominal voltage is slightly higher than S10K11 that results on a higher voltage clamping.

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
