# Peer review of "Dynamical Networks Modelling Applied to Low Voltage Lines with Nonlinear Filters"

_asi, doi:10.3390/asi3020018_

Round 1

Reviewer 1 Report

This work developed an iterative model to deal with  dynamical measurements  through a one-dimensional communication line model using  reflected and incident signals, which are dependent of the node parameters, proceeding then a time- step computation. 

However, the writing style of the paper has to be improved

The contribution and the motivation of the paper are not well formulated in the introduction. To the best of the knowledge, each introduction should at least point out a clear motivation by adding one or two relevant references inline with the paper. Which can be found in this paper The literature review cannot be found.  From line 61-70, the author basically mentioned 3 references which in my view are not exhaustive In addition, a separated section of the literature review should be made rather than a few sentences Also once the literature review is enhanced, section 4 of results & discussion should be enhanced accordingly.

Reviewer 2 Report

The author may give some references to support the discussion in the introduction.

The writing quality can be improved by using simple, plain languages. Also, it is better to put things in a straight manner rather than indirectly or using sentences in inverted order.

The author may consider articulate the main contributions at the beginning and then demonstrate the effectiveness and implications at the end. Currently, it may take some effort to judge the contributions from discussions.

Round 2

Reviewer 1 Report

The comments were addressed accordingly

Minor English check is required

Author Response

Dear Reviewer.

Much appreciated and very grateful for your suggestions and comments.

The English language was verified in the entire text, several spelling and grammar errors were corrected, at the best of our efforts.

Thanks.